# Photochemical diazidation of alkenes enabled by ligand-to-metal charge transfer and radical ligand transfer

Kang-Jie Bian[1,2], Shih-Chieh Kao[1,2], David Nemoto Jr. ●[1], Xiao-Wei Chen[1] & Julian G. West ●[1] ✉

Vicinal diamines are privileged synthetic motifs in chemistry due to their prevalence and powerful applications in bioactive molecules, pharmaceuticals, and ligand design for transition metals. With organic diazides being regarded as modular precursors to vicinal diamines, enormous efforts have been devoted to developing efficient strategies to access organic diazide generated from olefins, themselves common feedstock chemicals. However, state-of-the-art methods for alkene diazidation rely on the usage of corrosive and expensive oxidants or complicated electrochemical setups, significantly limiting the substrate tolerance and practicality of these methods on large scale. Toward overcoming these limitations, here we show a photochemical diazidation of alkenes via iron-mediated ligand-to-metal charge transfer (LMCT) and radical ligand transfer (RLT). Leveraging the merger of these two reaction manifolds, we utilize a stable, earth abundant, and inexpensive iron salt to function as both radical initiator and terminator. Mild conditions, broad alkene scope and amenability to continuous-flow chemistry rendering the transformation photocatalytic were demonstrated. Preliminary mechanistic studies support the radical nature of the cooperative process in the photochemical diazidation, revealing this approach to be a powerful means of olefin difunctionalization.

As one of the most prevalent structural motifs in bioactive molecules, pharmaceuticals, and molecular catalysts, vicinal diamines have intrigued medicinal and synthetic chemists for decades (Fig. 1a)[1]. Due to this central importance, continuous efforts have been devoted to accessing this useful moiety efficiently and directly. Of current vicinal diamine synthetic strategies, olefin diazidation stands out as an attractive approach as organic azide functional groups can be rapidly reduced to the corresponding amines, allowing for direct access to the diamine motif. Moreover, this protocol also confers unique synthetic advantages, including using olefins, a family of structurally diverse and abundant organic feedstocks, as modular and readily available starting materials. Additionally, the organic azide intermediates can be

developed into other valuable functionalities, including triazoles via 1,3-dipolar cycloaddition[2], imines via aza-wittig reaction[3], and other robust transformations, making them useful handholds in chemical biology and material science[4].

Early approaches to alkene diazidation are dependent on stoichiometric, highly oxidative oxidants and/or harsh reaction conditions such as high heat and strong acid, limiting the functional group tolerance of these transformations[5–8]. Further, many of these early methods are centered on the reaction of activated olefins such as styrenes, exhibiting low reactivity for unactivated alkyl olefins. Recent advances contributed by the groups of Greaney[9], Loh[10], Xu[11], Bao[12], Liu[13], and others have significantly expanded the substrate tolerance of olefin

[1]Department of Chemistry, Rice University, 6500 Main St, Houston, TX, USA. [2]These authors contributed equally: Kang-Jie Bian, Shih-Chieh Kao. ✉e-mail: jgwest@rice.edu

**a** Vicinal diamines in pharmaceuticals/natural product and synthesis

Biotin

U-50, 488
*k* agonist

Antitubercular agent

*Jacobsen* ligand

**b** Accessing vicinal diamines from diazidation of alkenes via outer-sphere single electron transfer (OSET) or electrochemical method (e-chem)

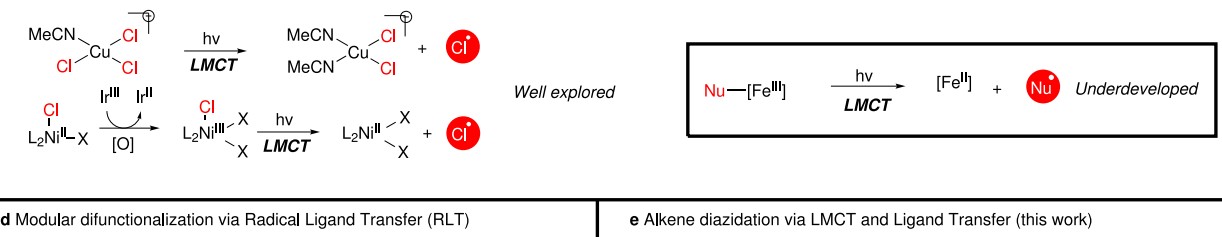

**c** Radical generation via Ligand-to-Metal Charge Transfer (LMCT)

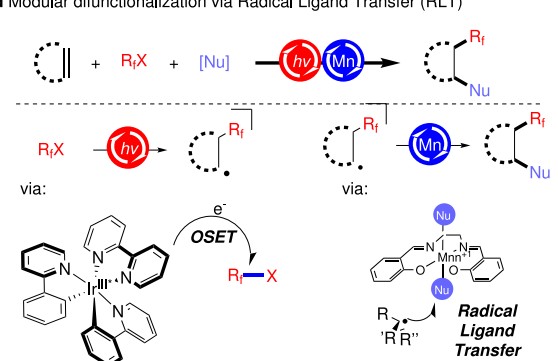

Well explored

Underdeveloped

**d** Modular difunctionalization via Radical Ligand Transfer (RLT)

via:

via:

*Radical Ligand Transfer*

**e** Alkene diazidation via LMCT and Ligand Transfer (this work)

*The merger of Ligand-to-Metal Charge Transfer and Radical Ligand Transfer*

*Azido Radical Generation (via ISET)*

*Radical Ligand Transfer*

**Fig. 1 | Background and project synopsis. a** Prevalence of diamine motifs in pharmaceuticals, natural products, and synthesis. **b** Previous works on azido radical generation via Outer-Sphere single electron transfer (OSET) pathway. **c** Ligand-to-Metal Charge Transfer (LMCT) enables radical generation. **d** Radical ligand transfer allows for facile delivery of orthogonal nucleophiles. **e** The synergistic cooperation of LMCT and RLT in alkene diazidation.

diazidation under thermal conditions, allowing for unactivated alkenes to be diazidated in high yield at moderate temperatures and without strongly acidic additives. Importantly, recent endeavors by Bao and coworkers have showcased the thermal, enantioselective diazidation of styrene-type alkenes using perester oxidants, providing a valuable tool for direct synthesis (with simple reduction) of chiral vicinal diamine[14]. While powerful, these approaches still require highly oxidizing, energetic, and expensive hypervalent iodine-derivatives or corrosive perbenzoate stoichiometric oxidants, presenting functional group compatibility concerns with oxidatively-labile substrates. Further, many of these methods require catalysts supported with complex ligand frameworks to function, presenting a barrier to the widespread

adoption of these methods. As an alternative to the traditional thermal chemical transformations, electrochemical methods have also offered a direct and appealing route to access these useful diazides motifs, with these methods garnering increasing interest in recent years due to their sustainability and high energy efficiency. Lin group reported an elegant electrochemical approach to diazidation of alkenes[15,16], exploiting the ability to achieve strong oxidative potentials at anodes in synergy with manganese[15] or aminoxyl[16] electrocatalysts to achieve dual azido group transfer onto alkenes. In a similar approach, efficient diazidation has been achieved with ppm loading of copper by Xu and coworkers, alleviating concerns of high catalyst loading in previous electrochemical diazidation[17]. While electrocatalysis has led to exciting advances in

olefin diazidation, the high complexity of electrochemical apparatuses and required multivariate optimization of factors such as electrode composition, morphology, and mass transport are current limitations of this approach. Taking these advances together, we imagined that a different, light-enabled mechanistic concept might allow us to eliminate the corrosive/expensive oxidants and/or complex electrochemical apparatuses previously needed in olefin diazidation and provide an efficient, easily accessible synthetic route to access vicinal diamine precursors with general functional group compatibility (Fig. 1b).

The continuing emergence of visible light-mediated photochemistry in modern organic syntheses has allowed for access to powerful, unconventional reaction manifolds to synthesize diverse small molecules[18–20]. Many recent methods have leveraged photoredox mechanisms, where photoactive complexes, often coordinatively-saturated mononuclear, expensive noble metal-based (especially Ru or Ir) species, perform bimolecular, outer-sphere single electron transfer (OSET) reactions to transiently generate organic radical species[21]. A complimentary strategy would be to perform inner-sphere single electron transfer (ISET), where the electron is transferred to -or from- a directly coordinated substrate. Light-driven ligand-to-metal charge transfer (LMCT) is one such ISET reaction, allowing for selective single electron oxidation of the ligand due to the required unimolecularity of the process[22]. Thus, LMCT of an anionic ligand results in a net homolysis of the metal-ligand bond, converting an anionic species into its radical form. Early studies from Kochi[23] in 1962 demonstrated the feasibility of LMCT to generate chlorine radical from cupric chloride upon irradiation and showed this reactive intermediate could be used to perform C–H chlorination and olefin dichlorination of simple hydrocarbons, a strategy later expanded by Sul'pin[24–26]. Following Kochi's early study of copper complexes, recent reports have expanded this reaction territory to other first-row transition metals such as nickel[27–30] and cobalt[31], taking advantage of the high degree of ligand-substitution lability of these elements to enable the dissociative pre-LMCT ligand-metal complex to readily form. Interestingly, unlike the well-demonstrated LMCT reactivity of copper-, nickel-, cobalt-based salts, iron salts have found fewer applications in photochemical transformations. It is possible that the short life time of the iron species might render it less capable to act as 'persistent radical' if radical ligand transfer occurs from a photoexcited state; however, iron-catalyzed radical ligand transfer reactions performed by cytochrome P450 and non-heme oxygenase enzymes and synthetic azide complexes do not require photoactivation, suggesting that this reactivity is accessible in the ground state. Limited examples of iron-LMCT have focused mostly on decarboxylation[32,33] or C–H functionalization via hydrogen-atom-transfer (HAT) processes[34]. Recent protocols deploying an iron-LMCT pathway have also shown success in generating alkoxy radicals from alkyl alcohols[35], halogen exchange processes of aryl halides[36] and in photochemical atom-transfer-radical-polymerization reaction[37]. However, there has been little exploration of using this approach to achieve alkene difunctionalization, despite it being one of the most direct ways to enhance molecular complexity.

Inspired by Kochi's original study[23] and detailed characterization of photoactive species in LMCT process of Cu$^{II}$ chlorocomplex (L$_n$CuCl$_3^-$) in acetonitrile by Mereshchenko[38], we posited that iron may be able to form an iron-azido complex in MeCN, affording 17e$^-$ Fe$^{III}$N$_3$ species, capable of diazidation of alkenes (Fig. 1c). Ideally, this intermediate could be formed directly from simple iron salts and a nucleophilic N$_3$ source, avoiding the need for expensive and complicated ancillary ligands. We envision the iron species could enable this reaction by merging the concepts of LMCT and ligand-transfer, having iron play dual roles as radical initiator and terminator. First, the azido radical could be generated from cheap, commercially available nucleophilic azide sources via Fe$^{III}$-N$_3$ homolysis (ISET) through LMCT. This azido radical could then add to the olefin substrate to generate a carbon-centered radical intermediate. Toward delivering the second

azide equivalent to the in situ generated carbon-centered radical, we hypothesized that we could take advantage of recent work we have disclosed[39] on bioinspired radical ligand-transfer (RLT), where a coordinated ligand on a metal center rebounds to an external radical intermediate to selectively deliver the coordinated functional group (Fig. 1d). We have found RLT to be a powerful means of diverting atom-transfer radical addition reactions and allow for incorporation of a diverse range of functionalities (including halogen, azide, and thiocyanate) in alkene difunctionalization reactions. The success of azide delivery in this manganese-mediated RLT process suggested to us that a similar step might be possible with iron.

Herein, we report a photochemical diazidation of alkenes using iron-mediated cascade ligand-to-metal charge transfer and radical ligand transfer (Fig. 1e). Aside from its unique mechanistic approach, this method has many advantages over previous technologies. First, iron is the most earth abundant metal and we have found this reaction to function using simple and cheap iron salts, providing significant sustainability and economic advantages for large-scale synthesis and pharmaceutical studies[40–42]. Second, our method does not require an added oxidant, ancillary ligand, or electrosynthetic apparatus, allowing for excellent functional group compatibility and simple reaction operation. Finally, in addition to batch conditions, continuous micro-flow reaction conditions have been shown to improve the mass-balance for our diazidation method and allow iron to perform catalytically in some cases. Together, this protocol presents a direct and attractive approach to access a broad range of organic diazides using only a simple iron salt and anionic azide source.

## Results

### Reaction design and optimization

We first set out to explore the possibility of this iron-mediated, photochemical diazidation by using the pent-4-en-1-yl benzoate as model substrate, nucleophilic trimethylsilyl azide (TMSN$_3$) as azide source, and different first-row transition metal salts under the irradiation of 427 nm blue light at room temperature (For more details, see supporting information). Detailed screening revealed that simple copper, cobalt, and manganese salts were not capable of converting nucleophilic azide into azide radical (Supplementary Table 1 and entry 2, Table 1). Indeed, in the case of copper salts, photoirradiation instead promotes LMCT of Cu–X to release halogen radicals, affording almost quantitative amounts of dihalogenated products in analogy to the findings of Kochi[23]. Having ruled out alternative metals, we next investigated iron salts to see whether an in situ formed Fe$^{III}$-N$_3$ is able to perform the azido radical generation needed for diazidation. Excitingly, we found that irradiating our olefin in the presence of 1.5 equivalents of Fe(NO$_3$)$_3$•9H$_2$O and 4 equivalents of TMSN$_3$ in acetonitrile afforded diazide product 1 in 84% isolated yield (entry 1, Table 1). Interestingly, other iron salts such as FeCl$_3$•6H$_2$O, Fe$_2$SO$_4$, or Fe(acac)$_3$ produced either dihalogenated product (entry 3, Table 1) for iron halides, similar to copper halide salts, or provide trace amount of diazidated product with a complex reaction mixture when using non-halide ligands such as acac (entry 4, Table 1). We hypothesize that the more strongly coordinating ligands in these species impede the coordination of acetonitrile and azide to generate our key LMCT precursor, preventing efficient reaction. We also observed a strong solvent effect on this reaction (Supplementary Table 2 and entries 5-8, Table 1); while DCM and THF were unable to promote the reaction (entries 5–6, Table 1), both acetone and ethyl acetate could provide diazidated product, albeit with lower yields (entries 7–8, Table 1). Further screening on concentration showed the reaction to be relatively insensitive to dilution and concentration (Supplementary Table 3 and entry 9, Table 1). Attempts to reduce the equivalents of azide source decreased the efficiency of the reaction (entry 10, Table 1). The equivalents of iron salt could be reduced to 1 equivalent to deliver the desired product in a comparative yield (entry 11, Table 1).

**Table 1 | Optimization of reaction conditions**

| Entry | Deviation from Standard Conditions | Yield (%)[a] |
|---|---|---|
| 1 | none | 86 (84) |
| 2[b] | 1.5 eq CuX₂ (X = Br, Cl) instead of Fe(NO₃)₃·9H₂O | ND |
| 3[c] | 1.5 eq FeCl₃.6H₂O instead of Fe(NO₃)₃.9H₂O | trace |
| 4 | 1.5 eq Fe(acac)₃ instead of Fe(NO₃)₃.9H₂O | trace |
| 5 | DCM | ND |
| 6 | THF | ND |
| 7 | EA | 64 |
| 8 | Acetone | 60 |
| 9[d] | 0.2 M, 0.05 M | 78, 82 |
| 10 | 2.0 eq TMSN₃ | 40 |
| 11 | 1.0 eq Fe(NO₃)₃·9H₂O | 80 |
| 12 | CFL (26 W), 390 nm, 456 nm, 525 nm LED (25 W) | 40–82 |
| 13 | no iron salt | NR |
| 14 | in the dark | NR |
| 15 | under air | 72 |
| 16[e] | under air with 20% Fe(NO₃)₃·9H₂O | 48 |

Reaction conditions: alkene (0.1 mmol, 1.0 equiv.), TMSN₃(4.0 equiv.), Fe salt (1.5 equiv.), and solvent (0.1 M), 24 h, RT, 427 nm Kessil blue LED (25 W). [a] $^1$H NMR yield is determined by using CH₂Br₂ as an internal standard. Isolated yield in the parentheses. [b] When used with CuCl₂ or CuBr₂, 40 and 68% of corresponding dihalogenated products were obtained. [c] 24% of dichlorinated product was obtained. [d] Similar yields were obtained, though reaction consistency was better with 0.1 M concentration. [e] Full consumption of starting material.

Control reactions revealed both iron and light irradiation are required, with full recovery of starting material in the absence of either, providing support for azido radical generation from a photo-induced ligand-to-metal charge transfer process (entries 12–14, Table 1).

Interestingly, we found this protocol is sufficiently robust to react without inert atmosphere (albeit with slightly lower yields), suggesting that sequestration of the in situ formed carbon-centered radical by RLT from an Fe$^{III}$-N₃ complex could be exceptionally fast, delivering second azide functionality onto aliphatic chain efficiently in the presence of competitive triplet oxygen (entry 15, Table 1). This robustness is remarkable compared to previous approaches and represents a significant advantage of our diazidation protocol.

**Scope of alkenes in diazidation (batch reaction)**

With optimized conditions in hand, we next sought to test the generality of this photochemical diazidation method. To our delight, a broad range of alkenes, including diverse unactivated (2-34) and activated (35-40) alkenes, could be transformed into corresponding diazidated products in moderate to excellent yields with 1.0–1.5 equivalents of iron salt under light irradiation (Fig. 2). First, simple aliphatic alkenes (2, 3, 4) and aryl rings bearing functional groups such as electron-donating methoxy (5) or electron-withdrawing chloro (6) and trifluoromethyl (7) all performed well, affording desired products in yields from 56–80%. Substrates bearing different protecting groups such as tosylate (8), N-Me-sulfonamide (9), and reductively labile 2,2,2,-trichloroethoxycarbonyl (10) also formed desired products efficiently, enabling subsequent synthetic elaboration of the diazidated molecules. Of particular note is that diverse alkenes bearing heterocycles such as N-phthalimide (11), N-methylpyrole (12) and pyran (13), all of which are tolerated in this system and give corresponding products in moderate to good yields. Moreover, acidic hydrogens such as

those found in sulfonamides, are also tolerated as demonstrated by product (14) being formed in 70% yield.

Next, we sought out to explore the generality of our protocol on different substrates bearing functionality that is sensitive to nucleophilic substitution or oxidation which are incompatible with conventional diazidation methods where strong and indiscriminate oxidants are often used. First, an unhindered primary bromide was tolerated by this reaction with no competitive azide displacement, forming diazide product (15) in 87% yield. The ester group (16) was also tolerated, providing corresponding product in good yield. Notably, functionalities that are susceptible to oxidative conditions such as sulfide (17), primary alcohol (18), ketone (19), aldehyde (20), and carboxylic acid (21) were all found to be compatible with our protocol, giving diazidated products in moderate to good yields, allowing for starting materials incompatible with previous, strongly-oxidizing methods to be engaged in diazidation. A series of polysubstituted alkenes were also studied to assess the scope of this reaction. 1,1-substituted (22, 23) and 1,2-disubstituted (24-26) alkenes provided expected products in good to high yields. Similarly, trisubstituted alkenes (19, 27, 28) also functioned well without significant decrease in yields. Next, the bulkier tetrasubstituted alkenes were also subjected under the standard conditions, affording corresponding products (29-31) in moderate to good yields. Taking these results together, this protocol offers a direct way to obtain functionalized tertiary aizdes which would be difficult to access via nucleophilic substitution using an azide nucleophile. Furthermore, our system was also compatible with cyclic alkenes, affording desired diazidated products (31-33) in 64–82% yields. It is worth mentioning that with two active olefinic motifs in the substrate, our method could provide single diazidation product (34) at the more electron-rich site chemoselectively. Lastly, we also conducted our diazidation protocol upon several prototypical activated alkenes,

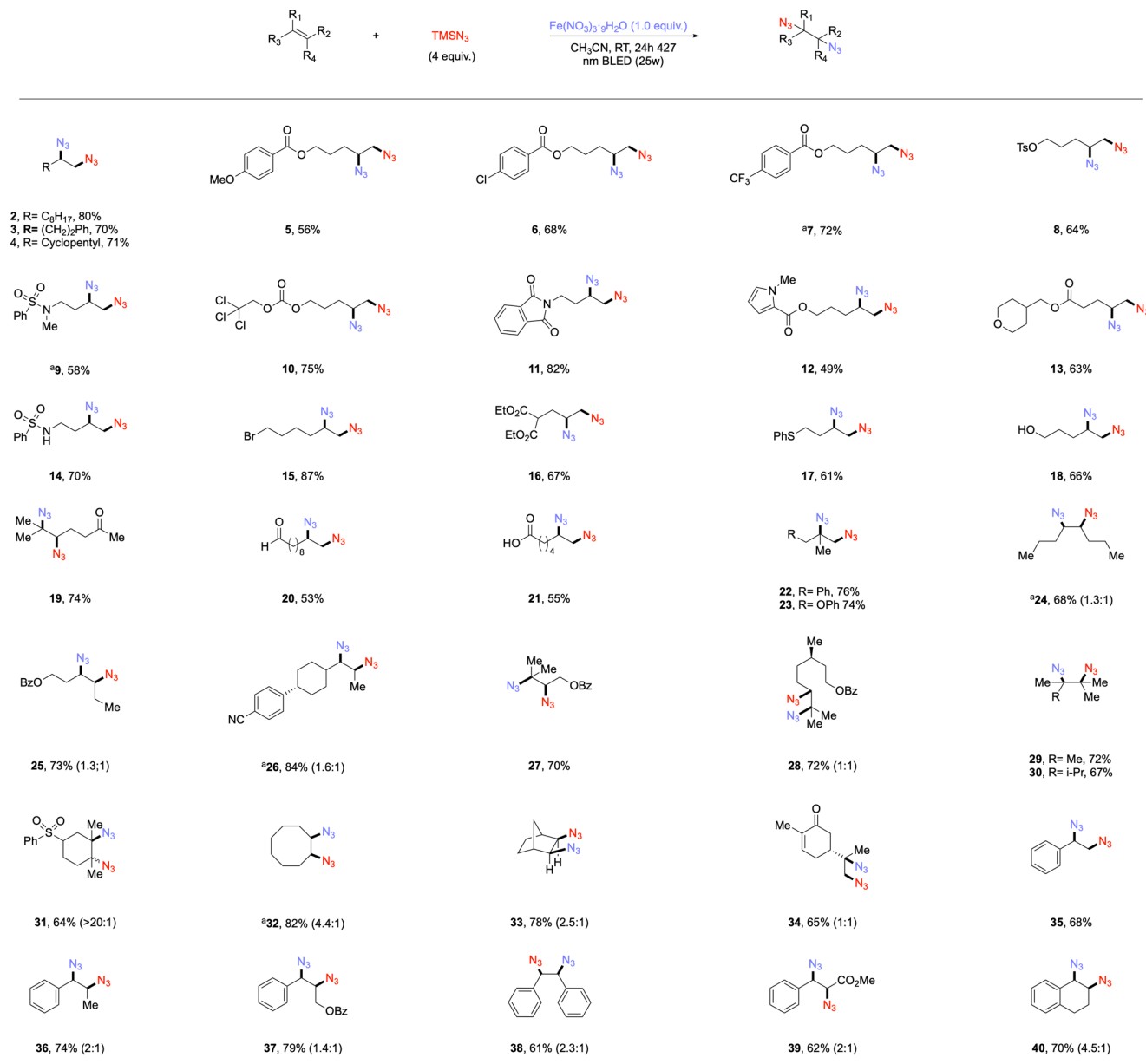

**Fig. 2 | Scope of diazidation of alkenes (batch reaction).** Reaction conditions: alkenes (0.1 mmol, 1 equiv.), TMSN₃ (4 equiv.), Fe(NO₃)₃.9H₂O (1.0 equiv.), MeCN (0.1 M), 24 h, RT, N₂, 427 nm Kessil blue LED (25 W). Ratios in the parentheses indicate the diastereomers proportion in corresponding products.

giving moderate to good yields of corresponding products (35–40). Compared with previous diazidation methods, our system shows unprecedented compatibility with diverse olefinic substrates, including oxidatively-labile substrates, under extremely mild conditions using exceedingly simple reagents.

## Scope of drugs/natural product-derived alkenes in diazidation (batch reaction)

Encouraged by high efficiency demonstrated of our diazidation protocol, we next endeavored to explore a different array of alkenes derived from commercially available active pharmaceutical ingredients (APIs) or natural products (Fig. 3). First, we derivatized several APIs including *Ibuprofen* (41), *Fluribiprofen* (42), *Loxoprofen* (43), *Isoxepac* (44), *Naproxen* (45), into their corresponding alkenes and performed the diazidation, affording moderate to good yields of diazidated products. Additionally, alkenes bearing natural product motifs were well-behaved in our system; *Flavone* (46), *L-Menthol* (47), *Borneol* (48), *Galactopyranose* (49) and *Glycyrrhetinic acid* (50) were all

compatible with this diazidation protocol, delivering corresponding products in good yields. Of note is that natural products such as *Olefic acid* (51) and *Mycophenolic acid* (52) bearing pre-existed olefinic moiety were also able to be diazidated in moderate to good yields, demonstrating practicality of our protocol in direct late-stage modification of bioactive molecules. The simplicity and generality of this method combined with the low cost, earth-abundance, and low toxicity of iron makes it ideally positioned for medicinal chemistry campaigns, allowing for the efficient and versatile synthesis of diverse organic diazides featuring biologically active motifs.

## Diazidation in 'flow'

Upon achieving highly efficient diazidation of diverse alkenes in batch with stoichiometric iron salt, we were intrigued by the possibility of developing a catalytic version of this protocol. However, when we carried out our standard reaction with 20 mol% of Fe(NO₃)₃.9H₂O under air, expecting aerobic oxygen to serve as the terminal oxidant for our process, we only observed ~50% of diazidated product with full

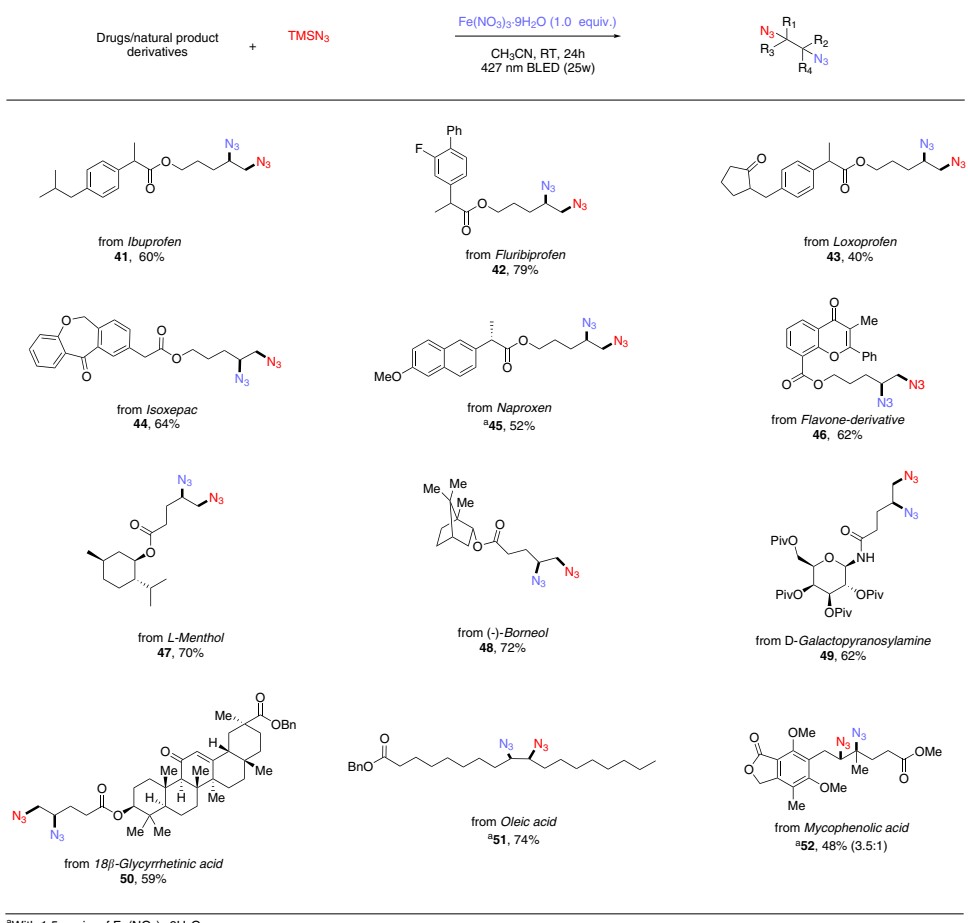

**Fig. 3 | Scope of diazidation of drugs/natural product-derived alkenes (batch reactions).** Reaction conditions: alkenes (0.1 mmol, 1 equiv.), TMSN$_3$ (4 equiv.), Fe(NO$_3$)$_3$·9H$_2$O (1.0 equiv.), MeCN (0.1 M), 24 h, RT, N$_2$, 427 nm Kessil blue LED (25 W).

consumption of starting material (entry 16, Table 1). While catalytic, this poor mass balance discouraged us from pursuing this batch condition further. Detailed screening of nitrogen/air mixtures or external chemical oxidants all rendered similar or lower yields (For more details, see supporting information). We posited that the low mass-balance could result from undesired intermediate sequestration due to poor reoxidation kinetics of iron in the batch reaction, where the in situ regenerated iron-azide species is not sufficiently concentrated to outcompete oxygen-related side pathways and thus not able to perform azido-ligand-transfer successfully.

Toward improving the mass balance of these substoichiometric iron reactions, we explored continuous-flow photolysis. Compared with batch photochemical reactions, the use of continuous-flow microreactors with photochemical methods has allowed significant improvement in reaction efficiency and surmounting of issues associated with batch photochemistry[43–48]. Due to better surface-area-to-volume ratios, the method can provide a uniform irradiation of entire reaction mixture, enabling shorter reaction time, large-scale syntheses, and improved mass-balance from less byproduct formation or substrates degradation. Hoping to capture these benefits, we performed a catalytic diazidation on several typical substrates from Figs. 2 and 3 with continuous-flow technique. We found comparable yields (up to 72%) and significantly improved (up to 92%) mass balance with these substrates using only 35 mol% iron, producing diazide products within only 135 minutes residence time (Fig. 4). Furthermore, performing diazidation of our optimizing substrate on 0.5 mmol scale produced diazide **1** in good yield and excellent mass balance. Although the absolute yields of these substrate are slightly lower than the batch

conditions, the significant acceleration of the reaction time and potential for large-scale process have demonstrated clear advantages of our photocatalytic flow diazidation approach.

## Derivatization, mechanistic studies, and possible mechanism

To demonstrate the synthetic utility of the versatile diazide products, we first performed the copper catalyzed azide-alkyne cycloaddition (CuAAC), affording 68% yield of cycloaddition product (Fig. 5, eq 1). Next, vicinal diamines could be synthesized via catalytic hydrogenation under 1 atm H$_2$ in high yields (Fig. 5, eq 2). Similarly, the diazide could also be reduced via Staudinger reduction followed by treatment with Boc$_2$O, furnishing protected vicinal diamine product in 79% yield (Fig. 5, eq 3). Bolstered by the wide scope and robustness of our method, we next endeavored to perform series of mechanistic studies to explore the elementary steps of this process (Fig. 5). First, the inclusion of 1 equivalent of radical scavenger 2,2,6,6-tetramethyl-1-piperidinyloxy (TEMPO) in the system, completely suppressed the reaction and alkene starting material was fully recovered, supporting the presence of radical intermediates in the reaction (Fig. 5, Panel B1). With the scavenger results in hand, we next sought to confirm the radical pathway and also explore the rate of eventual RLT process using two different radical clock substrates. N-tosylated-tethered diene furnished 5-exo-trig cyclization product in 60% yield (56), meanwhile, cyclopropyl-substituted alkene underwent facile ring-opening upon initial azide radical addition, giving expected product with excellent E/Z selectivity (57). As only rearrangement products and no 1,2-diazidated products were found in both entries, we postulate that iron-mediated, azido-ligand-transfer takes place after migrational ring-closing/opening, of which the rate would be slower than the rate

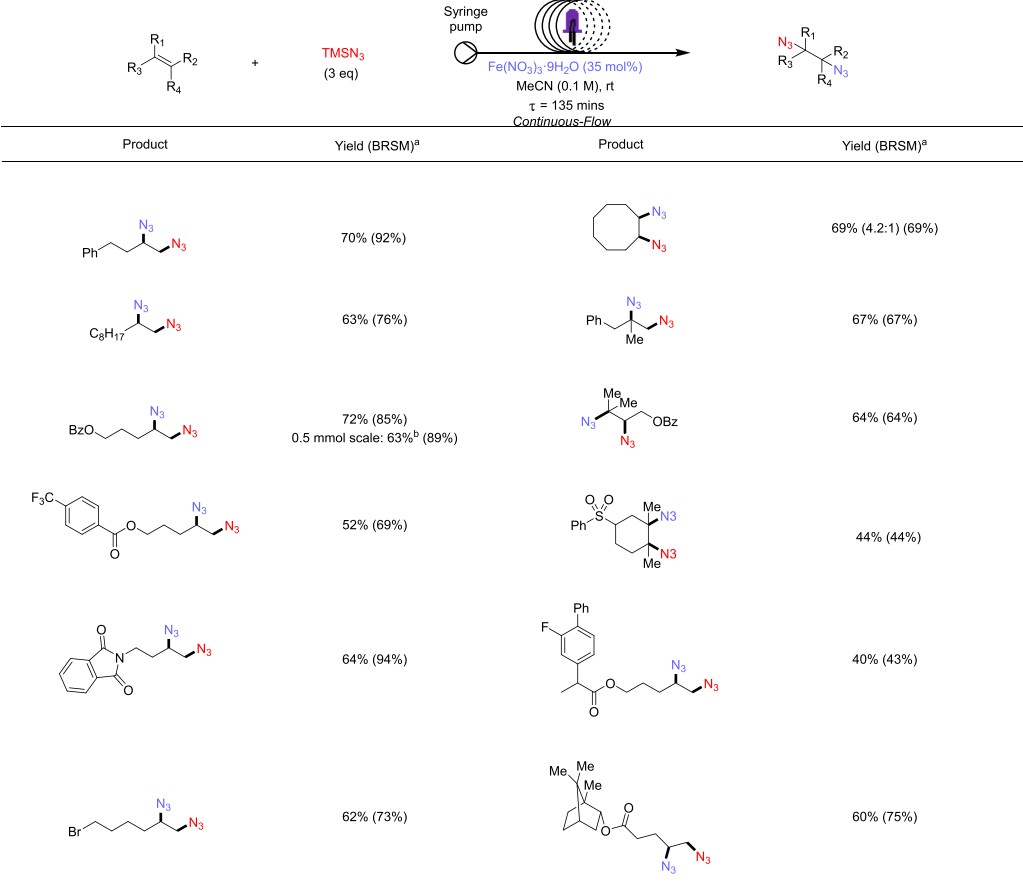

**Fig. 4 | Scope of diazidation of alkenes (flow reaction).** Reaction conditions: alkenes (0.1 mmol, 1 equiv.), TMSN$_3$ (3 equiv.), Fe(NO$_3$)$_3$.9H$_2$O (35 mol%), MeCN (0.1 M), 24 h, RT, under air, 2 × 390 nm Kessil blue LED (25 W).

constant of $2 \times 10^5 s^{-1}$ (approx. for 5-exo-trig) in our system (Fig. 5, Panel B2). Knowing the radical nature of the first azide initiation, we considered whether the second azide might be delivered via a radical-polar crossover (RPC) process, where the carbon-centered radical intermediate can be oxidized to a carbocation which can then be attacked by anionic azide. To test this pathway, we subjected 3,3-dimethylbut-1-ene to the standard conditions, knowing that the quaternary alkyl tert-butyl group is able to undergo '1,2-methyl shift' upon the generation of adjacent carbon cation. However, no migration was observed in this case, with 1,2-diazide (68) formed in 64% yield, suggesting RPC is less likely. This result is consistent with the second azide being delivered via RLT from an iron-azide species (Fig. 5, Panel B3). Next, we subjected 2-ethylnapthalene to our standard conditions, finding direct benzylic C–H azidation product (69), presumably via hydrogen-atom-transfer (HAT), to occur, albeit in lower yield, also suggesting the formation of a radical (e.g., •N$_3$) capable of HAT. In light of this finding, we next reacted a substrate that bears both an olefin and benzylic C–H bond to test the relative chemoselectivity of our system. To our delight, this experiment resulted in 70% of the diazidated product (70) with no C–H azidated product. To further showcase the chemoselectivity, we then subjected both the pilot substrate and 2-ethylnapthalene in one pot, where high yield of diazides was obtained from the alkene with only trace amount of benzylic azidation product. Indeed, both entries have demonstrated a high selectivity for diazidation over C–H azidation (Fig. 5, Panel B4).

Based on our collective mechanistic evidence and literature studies[11,22,49,50], we have proposed a possible pathway for iron-mediated photochemical diazidation. First, coordination of azide to dissolved iron salt produces Fe$^{III}$(N$_3$)$_x$(CH$_3$CN)$_y$, reminiscent of the 17e$^-$ Cu$^{II}$

chlorocomplex (L$_n$CuCl$_3^-$) in acetonitrile[38], which is capable of photo-induced LMCT, converting nucleophilic azide to its radical form (N$_3$·) (Step 1). The azido radical can then undergo radical addition to the alkene to provide a reactive carbon-centered radical intermediate (Step 2). Finally, this radical intermediate can be sequestered by another reactive iron-azide species, enabling facile azide delivery via a RLT process to provide diazidated products (Step 3). As iron is able to adopt a large number of oxidation states and spin states[51,52], it is postulated that such flexibility allows for the use of less than 2 equivalent of iron salt for the substrates reported in our protocol, as once azido radical is generated through LMCT, the lower-valent iron could undergo disproportionation and furnish higher-valent iron species that enables the latter azido-ligand-transfer[53,54].

In summary, we have demonstrated the photochemical diazidation of alkenes using earth abundant, cheap iron salts. This simple and general method allows for diazidation of broad range of alkenes by leveraging the merger of iron-mediated LMCT and radical ligand transfer. Mild conditions, excellent substrate scope tolerance and late-stage applications using biologically active molecules are all features of this protocol. The key to the success of this method is iron playing a dual role as radical initiator and terminator, where nucleophilic azide source is converted into its radical form via iron-mediated LMCT followed by azido radical addition onto a broad array of alkenes to produce a carbon-centered radical. This in situ generated radical can then be sequestered by another portion of iron-azide species via RLT to furnish diazide product. Future studies including catalytic protocol development and direct C–H functionalization based on synergistic cooperation of LMCT and RLT are ongoing in our lab.

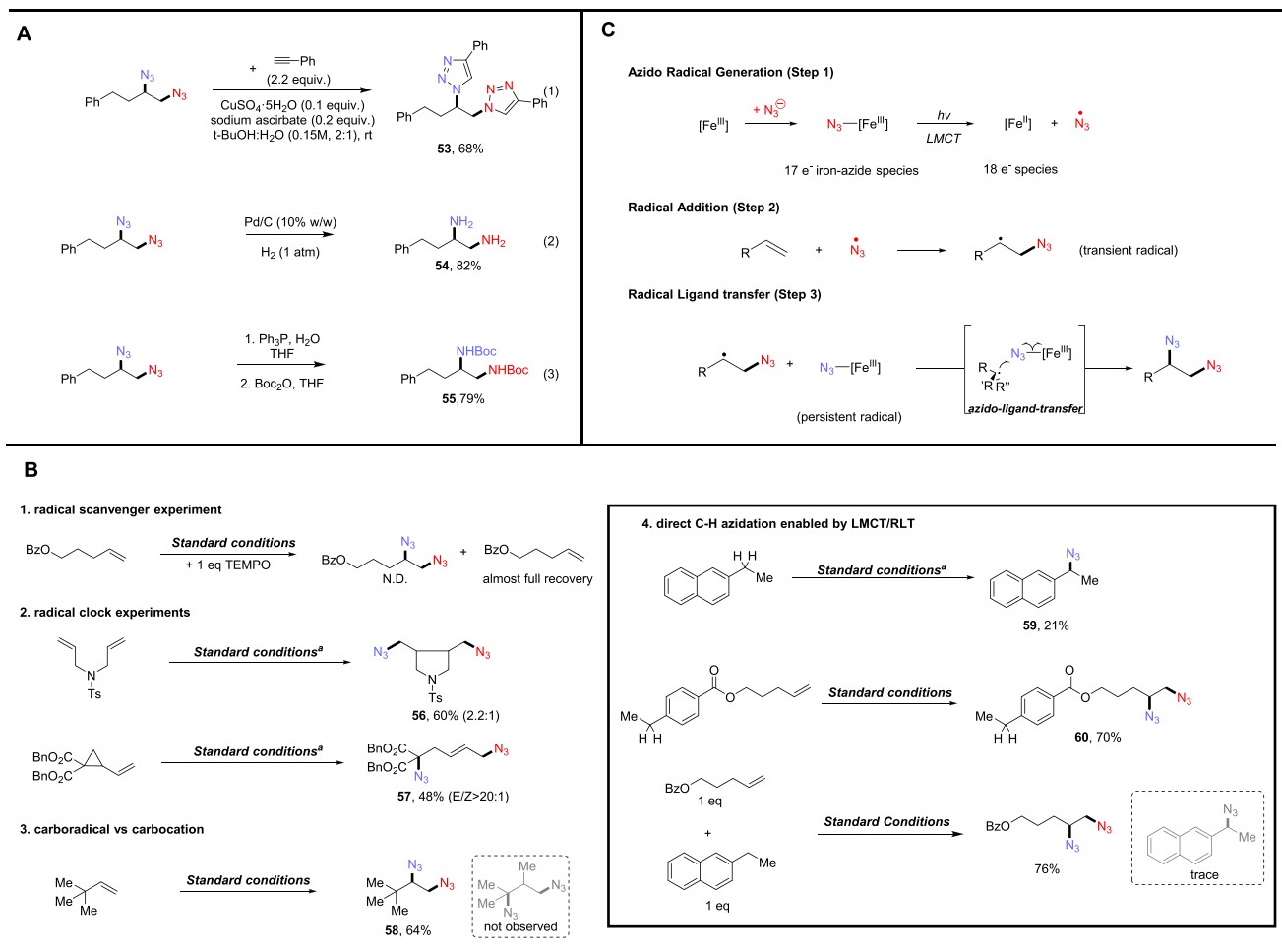

**Fig. 5 | Mechanistic studies and proposed pathway. A** Derivatization of diazides. **B** Mechanism studies. **C** Proposed pathway (azido radical generation → radical addition → radical ligand transfer).

## Methods

### General procedure for diazidation of alkenes

Fe salt (0.10 mmol, 1.0 equiv.) was added in an oven-dried 8-mL test vial containing a Teflon®-coated magnetic stir bar. The vial was evacuated and backfilled with $N_2$ (repeated for 4 times), followed by addition of alkenes (0.1 mmol, 1.0 equiv.) and $TMSN_3$ (0.40 mmol, 4.0 equiv.) in MeCN (1.0 mL, 0.1 M in regard to alkenes) via syringe under $N_2$. The reaction mixture was placed under 25 W 427 nm Kessil® light after sealing the punctured holes of the vial cap with vacuum grease and electric tape/parafilm for better air-tight protection and allowed to react at room temperature for 24 h. Following this, the reaction mixture was filtered through a pad of celite which was subsequently rinsed with DCM. The filtrate was concentrated, and the residue was then purified by flash column chromatography to give the corresponding diazidated products.

### General procedure for 'continuous-flow' diazidation of alkenes

Fe salt (0.035 mmol, 35 mol%), alkenes (0.10 mmol, 1.0 equiv.), TMSN3 (0.30 mmol, 3.0 equiv), MeCN (1.0 mL, 0.1 M in regard to alkenes) were added in an oven-dried 8-mL test vial containing a Teflon®-coated magnetic stir bar under N2. After the reaction mixture was withdrawn wtih a syringe, the syringe was connected to a FEP flow reactor and placed onto a syringe pump. the reaction mixture was pumped into a FEP flow reactor that was place under two 25 W 390 nm Kessil® light at a rate of 0.06 mL/h. (Note: the length of reaction tube in the flow diazidation scope is 30 cm, see below for more information). The reaction

mixture eluted from the outlet was discarded for the first 3 h and the subsequent portion was collected for another 16 h (on average ~1 mL). Following this, the collected portion was filtered through a pad of celite which was subsequently rinsed with DCM. The filtrate was concentrated, and the residue was then added dibromomethane as internal standard to determine the NMR yield of the corresponding diazidated products.

## Data availability

The authors declare that all the data supporting the findings of this research are available within the article and its supplementary information.

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

## Acknowledgements

J.G.W. acknowledge financial support from CPRIT (RR190025), NIH (R35GM142738), and the Welch Foundation (C-2085). J.G.W. is a CPRIT Scholar in Cancer Research. Dr. Yohannes H. Rezenom (TAMU/LBMS), Dr. Ian M Riddington (UT Austin Mass Spectrometry Facility), and Dr. Christopher L. Pennington (Rice University Mass Spectrometry Facility) are acknowledged for assistance with mass spectrometry analysis.

## Author contributions

K.-J.B. and S.-C.K. designed the project. K.-J.B., S.-C.K., D.N.Jr., and X.-W.C. performed the experiments. K.-J.B. and J.G.W. wrote the manuscript. J.G.W. directed the project. All authors interpreted the results in the manuscript.

## Competing interests

The authors declare no competing interests.
