## [Peer Review File · Nature Communications]

REVIEWER COMMENTS

Reviewer #1 (Remarks to the Author):

The synthesis of vicinal diazides under simple and mild conditions is attractive owing to the prevalence of vicinal diamines, the downstream products of vicinal diazides. In this manuscript, West and coworkers developed a visible light and iron mediated diazidation reaction of alkenes via a radical translation/radical addition/radical ligand transfer process. This reaction has the advantages of simple and general reaction conditions, good substrate tolerance, and greater cost efficiency. But meanwhile, the manuscript also has lots of defects, for instance:

1. This photoreaction has superior substrate tolerance in compare with previous diazidation methods, unactivated and activated alkenes bear diverse functional groups are all compatible. While this reviewer noticed that the tetra-substituted alkenes are not involved in the substrate scope, if this sterically alkene can approach the highly substituted di-tertiary azides?

2. In table 2, for compound 24-26 and 28-37, the meaning of the ratios in parentheses are not mentioned in both figure legend and main text.

3. This manuscript lacks of late-stage applications and downstream transformations of the vicinal diazide products. Vicinal diamines are prevalence in bioactive molecules, materials as well as ligands, while none of those vicinal diamines is obtained in this work.

4. There is no obvious advantage of the flow reaction. In Table 1, entry 16, the diazidation yield of pent-4-en-1-yl benzoate is 48% under air by using 20% Fe(NO₃)₃·9H₂O with batch reactor, while in Table 4, the yield is only 30% under the same reaction conditions with flow reactor, which means the flow reactor is not benefit for improving yield or accelerating reaction rate. Although the recovered starting material yield is up to 84% with flow reactor, but for a 0.1 mmol scale reaction, the recovery of starting material is unattractive.

That's more, the advantages of continuous-flow microreactor, large-scale reaction and shorter reaction time, are not illustrated by this reaction. The large-scale reaction with flow reactor should be achieved.

5. According to the flow setup picture in supporting information (Figure 2), the reaction system is placed under air, while Table 4 of the manuscript shows the reaction is under N₂ atmosphere.

6. In Figure 2, Panel A3, the benzylic C-H azidation product of 2-ethylnaphthalene is observed with only 21% yield. It is not strange that the substrate, bears a EWG group (-COOR), results in no C-H azidated product. Therefore, this experiment has no help for demonstrating the high selectivity.

7. The reaction mechanism is better to be supported by the DFT calculations.

This reviewer think this work is publishable in Nature Communications after considering all of the above revisions.

Reviewer #2 (Remarks to the Author):

The authors describe a protocol for a convenient method using an iron complex to synthesize a 1,2-diazide compounds from alkenes, which are equivalent of 1,2-diamines, common structural motifs in biologically active compounds. A few electrocatalytic methods reported recently (refs 14 and 15 in manuscript) showed a wide range of substrate scope, in which an azido radical is generated in situ. The authors argue that the complexity of the electrochemical reaction setup is high and the reaction scale is limited. The present reaction hopefully can improve on those shortcomings by using photoredox catalysis.

The scope is general with the reaction of sterically hindered internal olefins (19, 28) representing the advantage of the present radical pathway. The reaction shows tolerance with thiol (17) though iron has an affinity for sulfur. The reaction can take place in the presence of carboxylic acid without inducing a decarboxylation reaction. Yields are generally good, with minor issues.

The strength of this manuscript is that we always need other methods to achieve desirable transformations and the existence of the electrochemical catalytic diazidation reported by Lin and Xia (refs 14 and 15) should not heavily detract from this work. There is, however, a different precedent that is not cited and that is more problematic: Honglie Bao's ACIE 2021, 60, 12455-12460. This latter manuscript reports the Fe catalyzed diazidation of styrene type olefins and mirrors much of the reactivity reported in this paper with the caveat that it is catalytic in Fe and does not need light (this manuscript also reports a catalytic version of the reaction in flow but with more limited

scope and slightly depressed yields). Again, there are benefits to having orthogonal activation methods and this paper broadens the scope of the reaction beyond styrenes which is a plus.

The weakness of this manuscript is two-fold – this is a (mostly) stoichiometric Fe mediated reaction whereas the electrochemical pathways are catalytic and that is a major drawback. There are also some stylistic minor issues involving framing and renaming old concepts that should be addressed.

With respect to nomenclature: Photoredox catalysis recently developed intensively with Ru or Ir complex represents an outer-sphere single electron transfer (OSET). On the other hand, inner-sphere single electron transfer (ISET) is also known to include ligand-to-metal charge transfer (LMCT) in which an electron in an X-type ligand transfers into the metal center to form a radical species. The authors redefined such a phenomenon as “radical translation”. This is an unnecessary introduction of new nomenclature as the meaning is the same as the radical formation through LMCT – this should be changed. The authors claim the application of the iron-LMCT is limited to the decarboxylation or the C-H functionalization so far. But it has been applied to a HAT from an alkyl alcohol forming an alkoxy radical, a halogen exchanging reaction of aryl bromides, and polymerization (see: *Org. Lett.* 2021, 23, 21, 8413, *Org. Lett.* 2021, 23, 8413, *Macromolecules* 2017, 50, 20, 7967).

There are also some issues with the mechanism. The authors argue in the introduction that excited state Fe species have a short lifetime (which does not impact its ability to do LMCT and generate azido radical but does impact its ability to act as a “persistent radical” as claimed in the mechanism. There is evidence that ground state Fe azido species without photoactivation can act as radical acceptors to alkyl radicals (see Bao work above). Furthermore, many issues exist in the graphics. After ejection of azido radical the Fe complex is characterized as Fe(III) in Fig 1C and Fig 2B. This should be Fe(II). The authors also assign a negative charge to a putative Fe(N₃)₃L₃ complex in Fig 2B. This should be neutral with the caveat that they define L as being either CH₃CN or NO₃. Only the former qualifies as a typical L ligand.

On the whole, this is nice chemistry but not as transformative as the authors argue and with some issues in it. Prime among them for its suitability for Nature Comms is the strong precedent. I’m not convinced it is at the level where it should be for this journal but perhaps it can be made so.

Reviewer #3 (Remarks to the Author):

This manuscript by West and co-workers describes a photochemical iron-mediated radical diazidation of alkenes. The key finding is that iron-mediated ligand-to-metal-charge-transfer enables the generation of azido radical from Fe(III)-N₃ species to induce homolysis of C=C π bonds. After that, the resulting carbon-centered radicals could be successfully functionalized by another Fe(III)-N₃ species for the construction of the second C-N₃ bond.

Iron-catalyzed radical azidation of carbon-centered radicals have been well developed, even with some recent elegant asymmetric versions. In addition, diazidation of alkenes is a known transformation and the present work delivers the same overall reaction with similar substrate scope but with relatively low reaction efficiency—more than 20 mol % iron loading is generally required for the reaction. I think this is an interesting study that will be kind of interest to the synthetic community and I am convinced that this work will be well received and appreciated in a more specialized journal. However, I do not see the requirements met for publication in a top journal such as Nature Communications.

Additionally, In page 3, 'in a similar approach, efficient diazidation has been...by Xia and coworkers' should be '...by Xu and coworkers'. To date, this is the most efficient catalytic system for diazidation of alkenes. In my opinion, it is very easy to set up an electrolysis experiment with readily accessible electrodes and power supply in the lab and, anodic oxidation might be a good choice to help authors address the catalyst turnover problem in the present work.

Reviewer #1 (Remarks to the Author):

The synthesis of vicinal diazides under simple and mild conditions is attractive owing to the prevalence of vicinal diamines, the downstream products of vicinal diazides. In this manuscript, West and coworkers developed a visible light and iron mediated diazidation reaction of alkenes via a radical translation/radical addition/radical ligand transfer process. This reaction has the advantages of simple and general reaction conditions, good substrate tolerance, and greater cost efficiency.

We are grateful for the reviewer's enthusiasm for our work and we appreciate all the detailed suggestions given by the reviewer!

But meanwhile, the manuscript also has lots of defects, for instance:

1. This photoreaction has superior substrate tolerance in compare with previous diazidation methods, unactivated and activated alkenes bear diverse functional groups are all compatible. While this reviewer noticed that the tetra-substituted alkenes are not involved in the substrate scope, if this sterically alkene can approach the highly substituted di-tertiary azides?

We thank the reviewer for prompting us to further demonstrate the tolerance of alkenes substrates. In response to this question, we have complemented the scope with several examples of tetra-substituted alkenes. In general, these alkenes afford moderate to good yields of corresponding diazide products, further demonstrating the robustness of our protocol. The added examples are shown below and in the revised manuscript.

We have also added the following notes (highlighted in the manuscript):

Next, the bulkier tetrasubstituted alkenes were also subjected under the standard conditions, affording corresponding products (29-31) in moderate to good yields.

2. In table 2, for compound 24-26 and 28-37, the meaning of the ratios in parentheses are not mentioned in both figure legend and main text.

We thank the referee for bringing up this important point. The ratios in parentheses indicate the ratio of the diastereomers of these products, which are previously characterized in Ref.11-17. As reviewer suggested, we have added a footnote to the table 2 (shown below). We thank the referee for helping us to clarify this information.

Ratios in the parentheses indicate the diastereomers proportion in corresponding products.

3. This manuscript lacks of late-stage applications and downstream transformations of the vicinal diazide products. Vicinal diamines are prevalence in bioactive molecules, materials as well as ligands, while none of those vicinal diamines is obtained in this work.

We appreciate the constructional comments from the reviewer. As suggested, we have performed several late-stage/downstream transformations of the vicinal diazide products. The details are shown below:

Accompanying the scheme, we have also added following notes in the section “Derivatization, Mechanistic Studies and Possible Mechanism” of the revised manuscript (highlighted).

To demonstrate the synthetic utility of the versatile diazide products, we first performed the copper catalyzed azide-alkyne cycloaddition (CuAAC), affording 68% yield of cyclo-addition product (Figure 2, eq 1). Next, vicinal diamines could be synthesized via catalytic hydrogenation under 1 atm H_2 in high yields (Figure 2, eq 2). Similarly, the diazide could also be reduced via Staudinger reduction followed by treatment with Boc_2O , furnishing protected vicinal diamine product in 79% yield (Figure 2, eq 3).

4. There is no obvious advantage of the flow reaction. In Table 1, entry 16, the diazidation yield of pent-4-en-1-yl benzoate is 48% under air by using 20% $\text{Fe}(\text{NO}_3)_3 \cdot 9\text{H}_2\text{O}$ with batch reactor, while in Table 4, the yield is only 30% under the same reaction conditions with flow reactor, which means the flow reactor is not benefit for improving yield or accelerating reaction rate. Although the recovered starting material yield is up to 84% with flow reactor, but for a 0.1 mmol scale reaction, the recovery of starting material is unattractive.

We appreciate the referee for giving us this instruction. In this revision, we have re-optimized our flow technique and the new scope has shown significant improvement in yield (up to 72% absolute yield) and mass balance in comparison to previous manuscript using catalytic amount of $\text{Fe}(\text{NO}_3)_3 \cdot 9\text{H}_2\text{O}$ with the reaction time of 135 mins. This newly improved protocol has shown great tolerance towards diverse substrates and offered a complimentary method to the previous strategy for accessing diazides. The details are shown below:

That's more, the advantages of continuous-flow microreactor, large-scale reaction and shorter reaction time, are not illustrated by this reaction. The large-scale reaction with flow reactor should be achieved.

We appreciate this point from the referee and have also performed a larger scale reaction with the optimizing substrate, furnishing the expected product in comparable yield and good mass balance:

We have added this information in the revised flow table. In addition to the revised table, we have also revised the text description in the section of Diazidation in the 'Flow' of the revised manuscript as shown below:

We found comparable yields (up to 72%) and significantly improved (up to 92%) mass balance with these substrates using only 35 mol% iron, producing diazide products within only 135 minutes residence time. Furthermore, performing diazidation of our optimizing substrate on 0.5 mmol scale produced diazide **1** in good yield and excellent mass balance. Although the absolute yields of these substrate are slightly lower than the batch conditions, the significant acceleration of the reaction time and potential for large-scale process have demonstrated clear advantages of our photocatalytic flow diazidation approach.

5. According to the flow setup picture in supporting information (Figure 2), the reaction system is placed under air, while Table 4 of the manuscript shows the reaction is under N₂ atmosphere.

We thank the referee pointing out this error. We have revised the table 4 where shows the reaction is run under air. Moreover, the updated flow reaction setup and general procedures have been revised in the supporting information.

6. In Figure 2, Panel A3, the benzylic C-H azidation product of 2-ethylnaphthalene is observed with only 21% yield. It is not strange that the substrate, bears a EWG group (-COOR), results in no C-H azidated product. Therefore, this experiment has no help for demonstrating the high selectivity.

We appreciate the important comments from the reviewer. We agree that substituting the aryl with EWG could interfere with the reactivity of benzylic C-H bonds. Therefore, we have also added an intermolecular competition experiment where both unactivated alkene and ethyl naphthalene are added in one pot. The test showed almost exclusive chemoselectivity toward alkene diazidation, affording the corresponding product in high yields with only trace benzylic azidation.

We have added this new test to the revised scheme and have accompanied this revision with the following note (highlighted in the manuscript),

To further showcase the chemoselectivity of this reaction, we then performed an intermolecular competition experiment with both the optimizing substrate and 2-ethylnaphthalene, observing a high yield of diazide with only trace amount of benzylic azidation product. Together, these experiments demonstrate a high selectivity for diazidation over C–H azidation (Figure 2, Panel B4).

7. The reaction mechanism is better to be supported by the DFT calculations.

We thank the referee for this suggestion and appreciate the additional experiments and citations suggested by all reviewers that have given more support for our proposed mechanism. Our goal in this initial report is to demonstrate the key experimental aspects of this method, including its mild nature, high substrate tolerance, and light-mediated quality. Additionally, our improved flow photochemistry method demonstrates significantly improved yield, reaction time, and scalability compared to our initial submission, significantly advancing the state-of-the-art for iron-mediated diazidation.

We agree with this reviewer and the editor that expanding support for our proposed mechanism via DFT calculations is a promising direction for future studies and we are currently seeking out collaborators to pursue these investigations.

This reviewer think this work is publishable in Nature Communications after considering all of the above revisions.

We thank all the reviewers and the editor for their thoughtful comments that have allowed us to dramatically improve our manuscript. We have been the beneficiaries of a very thorough and constructive review process and are grateful for the reviewers' help to improve the scholarship and completeness of our manuscript!

Reviewer #2 (Remarks to the Author):

The authors describe a protocol for a convenient method using an iron complex to synthesize a 1,2-diazide compounds from alkenes, which are equivalent of 1,2-diamines, common structural motifs in biologically active compounds. A few electrocatalytic methods reported recently (refs 14 and 15 in manuscript) showed a wide range of substrate scope, in which an azido radical is generated in situ. The authors argue that the complexity of the electrochemical reaction setup is high and the reaction scale is limited. The present reaction hopefully can improve on those shortcomings by using photoredox catalysis.

We really appreciate the referee's recognition of the value of a simple photochemical diazidation method and their helpful suggestions to further improve this manuscript!

The scope is general with the reaction of sterically hindered internal olefins (19, 28) representing the advantage of the present radical pathway. The reaction shows tolerance with thiol (17) though iron has an affinity for sulfur. The reaction can take place in the presence of carboxylic acid without inducing a decarboxylation reaction. Yields are generally good, with minor issues.

The strength of this manuscript is that we always need other methods to achieve desirable transformations and the existence of the electrochemical catalytic diazidation reported by Lin and Xia (refs 14 and 15) should not heavily detract from this work.

We thank the referee for their thoughtful comments putting our work into context. We agree that there is high value in having a diversity of methods and reaction mechanisms to achieve important synthetic transformations. We believe this method complements the elegant work done by the Lin group and Xu group (Ref. 15-16 and Ref. 17), allowing for diazidation to be performed quickly and efficiently using common and cheap laboratory reagents and the only specialized equipment being a blue (batch) or purple (flow) LED light source. As suggested by the referees, we have rephased our wording in the manuscript to emphasize the importance of previous electrochemical strategies and the strength of our photochemical protocol:

As an alternative to the traditional thermal chemical transformations, electrochemical methods have also offered a direct and appealing route to access these useful diazides motifs, with these methods garnering increasing interest in recent years due to their sustainability and high energy efficiency.

[...]

In a similar approach, efficient diazidation has been achieved with ppm loading of copper by Xu and coworkers, alleviating concerns of high catalyst loading in previous electrochemical diazidation.¹⁷

There is, however, a different precedent that is not cited and that is more problematic: Honglie Bao's ACIE 2021, 60, 12455-12460. This latter manuscript reports the Fe catalyzed diazidation of styrene type olefins and mirrors much of the reactivity reported in this paper with the caveat that it is catalytic in Fe and does not need light (this manuscript also reports a catalytic version of the reaction in flow but with more limited scope and slightly depressed yields).

We are grateful to the reviewer for highlighting this important precedent and agree that this must be included in our discussion. As suggested, we have added the reference in the reference list and added the following notes in the manuscript (highlighted):

Importantly, recent endeavors by Bao and coworkers have showcased the thermal, enantioselective diazidation of styrene-type alkenes using perester oxidants, providing a valuable tool for direct synthesis (with simple reduction) of chiral vicinal diamine.¹⁴

Again, there are benefits to having orthogonal activation methods and this paper broadens the scope of the reaction beyond styrenes which is a plus.

We appreciate this important comment from the reviewer and agree it is important to have orthogonal activation methods to achieve the synthesis of high-value products, especially those with broad scopes that are not restricted to activated substrates. We believe that the photochemical activation mode of this reaction

compared with its broad scope reacting with both activated (e.g. styrenes) and unactivated alkenes under simple laboratory conditions will be of high interest and value to synthetic chemists!

The weakness of this manuscript is two-fold – this is a (mostly) stoichiometric Fe mediated reaction whereas the electrochemical pathways are catalytic and that is a major drawback.

We appreciate the suggestions from this referee and agree that a clear and convincing demonstration of catalysis will significantly increase the impact of our work. To achieve this, we have re-optimized our flow diazidation protocol in order to achieve a competitive complement to electrochemical approaches. In general, a wide array of alkenes (different functionalities, multisubstitution, drugs/natural product) can be transformed with catalytic amount of the iron salt into corresponding diazides in moderate to good yields, with good mass balance and a dramatically-shortened residence time of 135 mins. To our delight, this flow reaction can be readily scaled as well, proceeding on a 0.5 mmol scale with no modification of the reaction apparatus beyond a larger volume of reaction solution and correspondingly-scaled overall reaction time. We believe that this revised protocol offers an appealing alternative to both traditional thermal and electrochemical diazidation methods, representing a powerful photocatalytic diazidation approach. The details of the revised flow table are shown below,

Product	Yield (BRSM) ^a	Product	Yield (BRSM) ^a
	70% (92%)		69% (4.2:1) (69%)
	63% (76%)		67% (67%)
	72% (85%) 0.5 mmol scale: 63% ^b (89%)		64% (64%)
	52% (69%)		44% (44%)
	64% (94%)		40% (43%)
	62% (73%)		60% (75%)

To accompany the improved scope of our flow method, the following notes were added in the main text of the revised manuscript:

We found comparable yields (up to 72%) and significantly improved (up to 92%) mass balance with these substrates using only 35 mol% iron, producing diazide products within only 135 minutes residence time. Furthermore, performing diazidation of our optimizing substrate on 0.5 mmol scale produced diazide 1 in good yield and excellent mass balance. Although the absolute yields of these substrate are slightly lower than the batch conditions, the significant acceleration of the reaction time and potential for large-scale process have demonstrated clear advantages of our photocatalytic flow diazidation approach.

There are also some stylistic minor issues involving framing and renaming old concepts that should be addressed. With respect to nomenclature: Photoredox catalysis recently developed intensively with Ru or Ir complex represents an outer-sphere single electron transfer (OSET). On the other hand, inner-sphere single electron transfer (ISET) is also known to include ligand-to-metal charge transfer (LMCT) in which an electron in an X-type ligand transfers into the metal center to form a radical species. The authors redefined such a phenomenon as “radical translation”. This is an unnecessary introduction of new nomenclature as the meaning is the same as the radical formation through LMCT – this should be changed.

We appreciate these instructional comments from the reviewer. We agree that introducing new nomenclature is not necessary in this case and our process is well-described by LMCT and inner-sphere single electron transfer (ISET). As suggested, we have rephrased the introduction and the revised texts are shown below. Particularly, we have substituted the concept such as ‘radical translation’ with ‘radical generation through LMCT’ or ‘translating azide nucleophile into radical form’ with ‘converting azide nucleophile into radical form’.

Meanwhile, ‘radical translation and ligand transfer (RTLTL)’ in the figures has been revised to ‘The merger of Ligand-to-Metal Charge Transfer (LMCT) and Radical Ligand Transfer (RLT)’

For detailed rephrasing in the manuscript (highlighted), see below,

Title:

Photochemical Diazidation of Alkenes Enabled by Ligand-to-Metal Charge Transfer and Radical Ligand Transfer

Abstract:

Toward overcoming these limitations, we report the first photochemical diazidation of alkenes via iron-mediated ligand-to-metal charge transfer (LMCT) and radical ligand transfer (RLT). Leveraging the merger of these two reaction manifolds, we utilize a stable, earth abundant, and inexpensive iron salt to function as both radical initiator and terminator.

...

azide radical is generated from a cheap, commercially available nucleophilic azide source via $Fe^{III}-N_3$ photolysis.

...

Preliminary mechanistic studies support the radical nature of the cooperative process in the photochemical diazidation, revealing this approach to be a powerful means of olefin difunctionalization.

Introduction and optimization:

First, the azido radical could be generated from cheap, commercially available nucleophilic azide sources via Fe^{III}-N₃ homolysis (ISET) through LMCT.

[...]

Herein, we report the first photochemical diazidation of alkenes using iron-mediated cascade ligand-to-metal charge transfer and radical ligand transfer

Mechanistic studies:

it is posulated that such flexibility allows for the use of less than 2 equivalent of iron salt for the substrates reported in our protocol, as once azido radical is generated through LMCT ...

Conclusion:

leveraging the merger of iron-mediated LMCT and radical ligand transfer.

[...]

where nucleophilic azide source is converted into its radical form via iron-mediated LMCT followed by azido radical addition onto a broad array of alkenes to produce a carbon-centered radical.

For Fig 1e, the subtitle has been changed to 'The synergistic cooperation of LMCT and RLT in alkene diazidation.'

The authors claim the application of the iron-LMCT is limited to the decarboxylation or the C-H functionalization so far. But it has been applied to a HAT from an alkyl alcohol forming an alkoxy radical, a halogen exchanging reaction of aryl bromides, and polymerization (see: Org. Lett. 2021, 23, 21, 8413, Org. Lett. 2021, 23, 8413, Macromolecules 2017, 50, 20, 7967).

We thank the referee pointing out these important references. We have revised the manuscript and included these important examples in the reference list in addition to the following discussion in the main text:

Recent protocols deploying an iron-LMCT pathway have also shown success in generating alkoxy radicals from alkyl alcohol,³⁵ halogen exchange processes of aryl halides³⁶ and in photochemical atom-transfer-radical-polymerization (ATRP).³⁷

There are also some issues with the mechanism. The authors argue in the introduction that excited state Fe species have a short lifetime (which does not impact its ability to do LMCT and generate azido radical but does impact its ability to act as a "persistent radical" as claimed in the mechanism. There is evidence that ground state Fe azido species without photoactivation can act as radical acceptors to alkyl radicals (see Bao work above).

We appreciate for this referee pointing out this important concept. We agree that the shorter life time of excited state iron species could impact its ability to function as a persistent radical; however the beautiful work of Bao mentioned above and the known ability of iron to perform other radical ligand transfer reactions in the ground state without photoactivation (e.g. P450 hydroxyl transfer to alkyl radicals) suggest that this might not be necessary (i.e. light activation is only needed for the LMCT step, not the radical acceptor step).. We have revised our introduction to address both possibilities as helpfully pointed out by this reviewer.

...

It is possible that the short life time of the iron species might render it less capable to act as 'persistent radical' if radical ligand transfer occurs from a photoexcited state; however, iron-catalyzed radical ligand transfer reactions performed by cytochrome P450 and non-heme oxygenase enzymes and synthetic azide complexes do not require photoactivation, suggesting that this reactivity is accessible in the ground state.

...

Furthermore, many issues exist in the graphics. After ejection of azido radical the Fe complex is characterized as Fe(III) in Fig 1C and Fig 2B. This should be Fe(II). The authors also assign a negative charge to a putative Fe(N3)3L3 complex in Fig 2B. This should be neutral with the caveat that they define L as being either CH3CN or NO3. Only the former qualifies as a typical L ligand.

We are grateful for the referee pointing out this typo and providing these constructive suggestions! The figures have been corrected to show generation of Fe(II) after ejection of azido radical. We have also generalized the Fe(III) species shown in Fig 2B as we have not yet been able to determine the exact ancillary ligand environment of this species, which could be neutral or anionic as the referee correctly pointed out. The updated figures are included in the revised manuscript and shown below:

Figure 1C

Azido Radical Generation (Step 1)

Radical Addition (Step 2)

Radical Ligand transfer (Step 3)

Figure 2C

On the whole, this is nice chemistry but not as transformative as the authors argue and with some issues in it. Prime among them for its suitability for Nature Comms is the strong precedent. I'm not convinced it is at the level where it should be for this journal but perhaps it can be made so.

We are grateful for the detailed and constructive suggestions from this referee. The thoughtful feedback from all referees and the editor have allowed us to greatly improve our study and we believe this revised submission will be of great interest to broad readership of Nature Communications!

Reviewer #3 (Remarks to the Author):

This manuscript by West and co-workers describes a photochemical iron-mediated radical diazidation of alkenes. The key finding is that iron-mediated ligand-to-metal-charge-transfer enables the generation of azido radical from Fe(III)-N3 species to induce homolysis of C=C π bonds. After that, the resulting carbon-centered radicals could be successfully functionalized by another Fe(III)-N3 species for the construction of the second C-N3 bond.

Iron-catalyzed radical azidation of carbon-centered radicals have been well developed, even with some recent elegant asymmetric versions. In addition, diazidation of alkenes is a known transformation and the present work delivers the same overall reaction with similar substrate scope but with relatively low reaction efficiency—more than 20 mol % iron loading is generally required for the reaction. I think this is an interesting study that will be kind of interest to the synthetic community and I am convinced that this work

will be well received and appreciated in a more specialized journal. However, I do not see the requirements met for publication in a top journal such as Nature Communications.

We appreciate the reviewer's commitment to help us improve our manuscript and their recognition of the interest of our photochemical diazidation strategy.

Although the diazidation of alkenes has been developed with both thermal and electrochemical approaches, we agree with reviewers 1 and 2 that a complementary photochemical activation strategy proceeding via a new tandem LMCT/RLT mechanism is of great value and interest to a broad chemical audience. Further, the simplicity of this reaction system (proceeding using simple, commercial, and cheap reagents with no need for complex ligands or reaction apparatuses) combined with the broad substrate scope of this approach further increases the impact of this work and establishes it as an inseparable complement to the previous elegant chemical and electrochemical methods.

In addition to the interest of the work, we are grateful for the insightful feedback of all reviewers and the editor that has allowed us to greatly improve the scholarship, completeness, and impact of our revised manuscript. In particular, we have re-optimized our flow diazidation protocol and achieved significant improvement, providing good yields and mass balance of diverse diazides within only 135 mins. Importantly, we also demonstrated the flow reaction to be scalable, indicating the practicability of our protocol. On the advice of all referees and editor, our revised flow diazidation scope is included in the revised manuscript and below:

Additionally, a major contribution of this work is not just the first photochemical diazidation of alkenes (which complements chemical and electrochemical approaches), but also introducing a new mechanistic approach where iron-mediated LMCT is utilized in tandem with radical ligand transfer (RLT) to achieve alkene difunctionalization, where iron plays a dual role as both radical initiator and terminator. We believe that this approach will be general for many difficult-to-achieve olefin functionalizations and other radical reactions.

Taken these points together, we believe our first photochemical diazidation of alkenes should be of high importance to the broad readership of Nature Communications and inspire further work in this promising area.

Additionally, In page 3, ‘in a similar approach, efficient diazidation has been...by Xia and coworkers’ should be ‘...by Xu and coworkers’. To date, this is the most efficient catalytic system for diazidation of alkenes. In my opinion, it is very easy to set up an electrolysis experiment with readily accessible electrodes and power supply in the lab and, anodic oxidation might be a good choice to help authors address the catalyst turnover problem in the present work.

We are grateful to the referee for pointing out this error and as suggested we have corrected the manuscript to credit the authors of this impactful study. To further highlight this important work, we have added the following notes in the revised manuscript,

In a similar approach, efficient diazidation has been achieved with ppm loading of copper by Xu and coworkers, alleviating concerns of high catalyst loading in previous electrochemical diazidation.

Indeed, we agree that this recent example is by far the most efficient catalytic system for alkene diazidation. In terms of catalyst turnover. The extremely broad substrate scope and clean reaction system have similarly showcased the importance of this work to our community.

While this copper electrocatalytic approach has many advantages in terms of reaction efficiency and substrate scope, we will stress that there is high value in developing methods with complementary or orthogonal activation modes to allow for functional group compatibility issues to be overcome and maximize the accessibility of desirable transformations by not relying on complex and/or expensive reaction apparatuses. Moreover, in comparison to expensive and complex electrolysis setups (eg. IKA electrosyn is listed price at \$2500+ USD) and sometimes limited scalability in electrochemical transformations due to mass transport and electrode geometry constraints, our photochemical method only requires simple iron salts that costs as low as \$0.07 per gram in standard reaction glassware with blue LED light, a condition that most synthetic labs in academia and industry can readily achieve.

Further, our revised flow protocol also shows that the diazidation can be rendered catalytically with much greater mass efficiency and it behaves well on large scale, presenting a much more synthetically-attractive catalytic variation of our method. Moreover, the flow chemistry apparatus can be readily (And cheaply) prepared from common laboratory instruments (syringe pump, LED light source, and syringe) and cheap supplies (FEB tubing and a luer adapter), making this condition accessible to those wanting to perform the reaction catalytically. We think this new insight where the reactive radical intermediate can easily be generated through iron-mediated LMCT, an important type of transformation under the manifold of inner-sphere single electron transfer (ISET) and the first application of iron LMCT in alkene difunctionalization would be of great interest to the readership of Nature Communication and we hope it will inspire the chemistry community to access diverse radical species (other than azido radical) from exciting new applications of this approach.

REVIEWERS' COMMENTS

Reviewer #1 (Remarks to the Author):

Julian and coworkers have made a good contribution in the photochemical diazidation of alkenes via a novel ligand to metal charge transfer process. The authors have done a good revision and almost all the concerns of three reviewers have been well addressed.

As a similar manuscript has been accepted by the office, this reviewer thinks the editorial office should not pay attention to the third reviewer's negative comments and should accept this manuscript directly without any delay. That would be fair.

Reviewer #2 (Remarks to the Author):

The manuscript has been suitably revised. I think it is acceptable for Nature Comms.

Reviewer #1 (Remarks to the Author):

Julian and coworkers have made a good contribution in the photochemical diazidation of alkenes via a novel ligand to metal charge transfer process. The authors have done a good revision and almost all the concerns of three reviewers have been well addressed.

As a similar manuscript has been accepted by the office, this reviewer thinks the editorial office should not pay attention to the third reviewer's negative comments and should accept this manuscript directly without any delay. That would be fair.

We are grateful for the reviewer's enthusiasm for our work and their helpful suggestions through the review process to improve our study!

Reviewer #2 (Remarks to the Author):

The manuscript has been suitably revised. I think it is acceptable for Nature Comms.

We are similarly grateful for this reviewer's enthusiasm for our work and their helpful suggestions through the review process. These comments allowed us to dramatically improve our study!